# Explaining Global Turkey Biometric Diversity Through Principal Component Analysis

**DOI:** 10.3390/ani15172537

**Published:** 2025-08-28

**Authors:** José Ignacio Salgado Pardo, Antonio González Ariza, Laura Carranco Medina, José Manuel León Jurado, Juan Vicente Delgado Bermejo, Stefano Paolo Marelli, Silvia Cerolini, Luisa Zaniboni, María Esperanza Camacho Vallejo

**Affiliations:** 1Department of Genetics, Faculty of Veterinary Sciences, University of Córdoba, 14071 Córdoba, Spain; josalgadopardo@outlook.com (J.I.S.P.); lauracarrancomedina3e@gmail.com (L.C.M.); juanviagr218@gmail.com (J.V.D.B.); 2Agropecuary Provincial Centre, Diputación de Córdoba, 14071 Córdoba, Spain; jomalejur@yahoo.es; 3Department of Veterinary Medicine, University of Milan, 26900 Lodi, Italy; stefano.marelli@unimi.it (S.P.M.); silvia.cerolini@unimi.it (S.C.); luisa.zaniboni@unimi.it (L.Z.); 4Department of Agriculture and Ecological Husbandry, Area of Agriculture and Environment, Andalusian Institute of Agricultural and Fisheries Research and Training (IFAPA), Alameda del Obispo, 14004 Córdoba, Spain; mariae.camacho@juntadeandalucia.es

**Keywords:** phenomics, *Meleagris gallopavo*, breed diagnosis, meta-analysis, zoometrics

## Abstract

This study examined the morphological diversity of domestic turkeys by analyzing data from 97 reports across 28 studies covering 15 breeds. The researchers analyzed body measurements and found that certain traits, such as leg length, body mass, and tarsus size, were important in distinguishing between breeds, particularly among females. Notable differences in head and leg features were also observed between males and females. The analysis revealed two main regional groups of turkey: the African and the Mediterranean. This study emphasizes the importance of standardized methods for describing turkey breeds and provides valuable guidance for future research.

## 1. Introduction

Biodiversity can be measured by the genetic variability that explains the different morphologies, behaviors, or adaptations within a species [1]. In the case of domestic birds, there is a huge range of phenotypic diversity, many times greater than between different wild species [2]. However, in the case of the turkey, this phenotypic diversity is not as evident, especially when compared to other poultry species such as chickens [3]. Domesticated in the current Mexico around the second century AD [4], turkeys arrived in the port of Seville at the beginning of the XVI century [5]. In a matter of a few years, they reached almost every country in Europe [6], as well as many other parts of the world. This was due to the excellent quality of its meat, together with the lack of religious restrictions and its great heat tolerance and adaptability [7]. Therefore, its relatively recent domestication [8], together with its rapid geographical expansion [3,5], could explain the lack of phenotypic diversity compared to other domestic birds.

As domesticated turkeys spread, the populations involved began to differentiate. Adaptation to the new environments and the new roles they played in the different communities and cultures, together with the genetic drift and isolation, led to the great variability of the populations currently available [9,10]. What is more, 187 turkey landraces have been described worldwide in the Domestic Animal Diversity Information System (DAD-IS) database of the Food and Agriculture Organization of the United Nations (FAO) [11]. However, it barely represents 5% of the total number of poultry breeds described, and turkeys stand under the chickens, ducks, and geese, with 2530, 422, and 262 described breeds, respectively. Furthermore, the knowledge of the current status of turkey breeds is still underdeveloped. This is evidenced by the scarce populations described in many continents and the unknown status of 65% of the populations described, as shown in Figure 1.

Defined as ‘a group of homogeneous animals distinguishable from others’ [12], breeds should undergo a comprehensive description process. This process of describing these differentiating traits, usually referred to as characterization, is essential for assessing the diversity of animal genetic resources (AnGR) [9]. The characterization of AnGR includes historical, genomic, and phenomic studies, among which the latter includes the identification of their external and production characteristics [12]. Morphological studies have been widely employed as the first approach to characterize AnGR before molecular genetic studies [13]. It is due to its ability to classify and distinguish within and between populations based on their appearance similarities and differences through the implementation of statistical analyses. The importance of developing these studies according to a homogeneous criterion in terms of the measurement protocol and their anatomical references is crucial to manage breed diversity worldwide [12]. For this aim, the FAO Food and Agriculture Organization of the United Nations published specific guidelines for the morphological characterization of AnGR in 2012 [12].

The implications of the morphologic information go beyond the description of populations. Zoometry allows the definition of breed standards, which serves to maintain specific phenotypes characteristic of each population [12]. Breed patterns help breeders’ associations maintain purebred lines by avoiding genetic erosion of other genotypes through simple, visual measurements that are easy to record. Therefore, this is a crucial step in the correct management of endangered populations [13]. On the other hand, morphological information is also crucial for the advanced selection programs of well-established breeds. The linear assessment is a selection criterion that exploits the correlations between a particular morphology and a specific trait under selection [14,15,16]. Additionally, some zoometric measurements are associated with the productive life of animals, and are also recorded to enhance long-term productivity [17].

In this respect, several studies have recently been published addressing the genetic characterization of turkey landraces from many parts of the world [13,18,19,20,21,22,23]. Most of the articles mention the FAO’s guidelines in the experimental design and measures. However, chickens are the only poultry species included in the guidelines, and the extension of the same measures described for chickens to other domestic birds is suggested [12]. However, this may not be appropriate, due to the anatomical peculiarities of each species. The turkey meets an additional difficulty, since there is a marked lack of knowledge about the species. For example, while there is a study in chickens addressing the morphological variance between populations [24], no similar studies have been developed in turkeys. This is evidenced in the growing but still scarce literature on breed characterization in the species.

Therefore, the present study aims to analyze the morphological variance between the worldwide turkey landraces through the use of multivariate statistical analysis. This would be the first study to address this issue, as no similar research is available on the domestic turkey. This would lead to a deeper knowledge of the current diversity available, as well as to determine similarities and differences between the populations from whose information is available. In addition, the results of the present work would help to determine the most variable parameters between populations and the correlation between them. This work would be of great interest to breeders’ associations and institutions, as well as to the international poultry industry. The results obtained might help design future morphological characterization studies in the species, and could lead to the development of measuring protocols intended for conserving and improving the domestic turkey populations worldwide.

## 2. Materials and Methods

### 2.1. Data Collection

To develop a comprehensive analysis and to include as much information as possible, the data collection protocol was based on previous meta-analyses developed in the livestock and poultry fields [25,26]. According to the literature, the repositories scholar.google.es (accessed on 25 July 2025) and www.sciencedirect.com (accessed on 25 July 2025) were consulted from January to June 2024. As described in the bibliography above, other repositories such as https://pubmed.ncbi.nlm.nih.gov/ (accessed on 25 July 2025) or www.webofscience.com (accessed on 25 July 2025) were discarded due to a lack of data extraction tools available on these websites. In the case of articles with reading restrictions, access has been granted by the University of Córdoba’s library service.

The following keywords were used in the search for the papers providing the data for the analysis: “morphological/morphometric characterization” followed by “turkey”, “*Meleagris gallopavo*”, “landraces/breeds/genotypes”, and other semantically related terms, as described in the literature [27]. Only papers published in English and Spanish were considered, and no limit on the date of publication was defined. Following the aforementioned directives, a total of 28 papers dealing with the morphological characterization of turkey landraces were found. These studies were published between 1956 and 2023 and written in English and Spanish. From this point, each report of a given sex, breed, and variety (if reported) within an article was considered an individual observation to generate the database on which the analyses were performed. Therefore, a total of 97 observations (49 from males and 48 from females) of animals between 4 and 11 months of age (when each genotype reached adult weight) were collected. Observations included up to 15 different turkey breeds, which are described in Table 1. Articles reporting observations with combined male and female means were discarded. This is firstly due to the discrepancy with observations from sex-segregating articles, and secondly because of the inadequacy of considering both sexes together, given the remarkable sexual dimorphism of the species [26,28].

Each observation included at least one of the following parameters: body weight, wingspan, body length, skull length, skull width, beak length, beak width, snood length, neck length, breast circumference, keel length, thigh length, tarsus length, tarsus depth (anteroposterior diameter), tarsus width (lateral diameter), middle toe length and, only for males, beard length. Other measures, such as eye length and width, snood width, dorsal length, and tail length, were also reported in papers but were excluded from the analysis as they were described in only one observation each.

To avoid estimation errors, the units employed in every research paper were standardized. In addition, the measurement protocol was revised in each article to ensure that all parameters were taken using the same anatomical references. The few research articles that did not describe the measurement protocol in detail were included if they referred to the FAO’s guidelines for morphological characterization [12]. Some of the morphometric parameters collected were then used to compute seven biometric indices, following the formulae described in the literature [10,13,40,47]. Finally, although qualitative variables (morpho-structural and cutaneous, also called ‘phaneroptic’) were recorded as well, they were discarded from the analysis due to the high variability of inclusion criteria and nomenclature. Table 2 exhibits the body measurements and the biometric indices computed in the present study, alongside e their description and the abbreviation that will be used in further sections.

### 2.2. Ethics Statement

No ethical review was needed since the data sources were already published studies; no new animals were involved in the data collection.

### 2.3. Data Analysis

Data collection and processing, as well as some of the graphics generated, were carried out using the software Microsoft Excel 2018 (Microsoft Corporation, Redmond, WA, USA). The different principal component analyses (PCAs) were performed with the Principal Component Analysis package of XLSTAT software (Addinsoft Pearson Edition 2014, Addinsoft, Paris, France).

Three PCA approaches were performed, always with the breed as the dependent variable, and separately for each sex. What varied between the different PCAs carried out was the set of explanatory variables taken into account in each analysis. In this way, an initial analysis was performed, taking all the biometric measures and indices together. From this initial analysis, the Pearson correlation matrix automatically generated in the PCA routine was used to create a heat map through the website http://heatmapper.ca/ (accessed on 25 July 2025). The Cohen correlation magnitude (<0.1 no effect, 0.1–0.3 weak, 0.3–0.5 median, and >0.5 strong correlations) was used to the interpretation of the values correlation magnitude [48]. A second PCA was then performed using the seven biometric indices as the only explanatory variables. The two principal components (PCs) of this analysis were used to plot the observations territorially by sex using the ggplot2 package in RStudio^®^. Finally, the body measures were grouped into three groups for the third and final approach to the analysis. These regions were ‘head’ (including skull length, skull width, beak length, beak width, and snood length), ‘trunk’ (including body weight, wingspan, body length, neck length, breast circumference, and keel length), and ‘legs’ (including thigh length, tarsus length, tarsus depth, tarsus width, middle toe length and beard length for males). A PCA was then performed for each body region and, once more, separately for each sex, to analyze the different variance-explaining potential of each corporal region.

## 3. Results

### 3.1. Descriptive Statistics

The means and standard deviations of the different parameters in the breeds included are represented in Table 3 for males and Table 4 for females. A great variability in morphometric traits is evidenced in both sexes, especially for body weight, in which a difference of three times the value can be drawn between the heaviest (Commercial) and lightest (Guatemalan) breeds. In both sexes, the Commercial genotype has the highest values for almost all the variables included, although for some traits, such as body and tarsus length, no great differences with other genotypes are observed. A marked sexual dimorphism is present in every breed included; however, the degree of differences between sexes varies with breed. The effect of sexual dimorphism on the morphometric parameters can be consulted in the Appendix A. Lastly, the mean values and standard deviation (if more than one report per genotype) of the biometric indices are depicted for both sexes in Table 5.

### 3.2. Correlation Matrix Among Explanatory Variables

Pearson’s correlation matrix generated in the first PCA routine, including all the morphometric parameters and biometric indices, was depicted as a heatmap for each sex, which is represented in Figure 2. Moderate (0.3–0.5) to strong (>0.5) positive and negative correlations were found between the different morphometric parameters included in the analysis, particularly in females. In males, stronger correlations were found between weight and tarsus depth (+0.876), beak width and beard length (+0.853), and tarsus length and middle toe length (+0.780). On the other hand, the stronger correlations exhibited in females were those between tarsus depth and width (+0.919), followed by keel length and middle toe length (+0.886), and between beak width and tarsus width (−0.671).

### 3.3. Principal Components Analysis and Model Generated

The first approach of PCA was performed using all the parameters (morphometric measures and biometric indices) of the present study, separately for each sex. However, due to the great number of variables considered (24 for males and 23 for females), an extremely complex model was generated (Appendix A). Moreover, the explanatory cumulative variance of the first two PCs did not achieve 70% in any of the sexes, which is the generally accepted cut-off point [49]. Due to the complexity of the model and the lack of variance-explanatory properties of the first principal component (PC1), a second analysis was carried out employing the biometric indices as the only explanatory variables. Despite including fewer variables, it showed greater explanatory properties since 70% of the explanatory variance was exceeded by combining the first two PC in both sexes (Figure 3). Moreover, 50% of the explanatory variance was covered by the PC1 in the case of males.

Table 6 contains the squared loadings of each biometric index for all the PCs generated in both sexes. The squared loadings represent the proportion of the variance of the variables explained by the components [50]. Therefore, variables exhibiting increased squared loadings would be those most contributing to the variance explanation for each PC [51]. Massiveness, long-leggedness, and body mass index were the variables exhibiting the greatest explanatory variance contribution in the PC1 in males. The same trend is found in females, with the addition of the shape index in PC2. This reinforces the greater relative importance of PC1 in the case of the male model compared to females, since this PC showed the highest explanatory performance of the majority of variables. On the other hand, in the female model, greater cumulative explanatory power was achieved compared with males when combining the first three PCs.

All observations were graphically represented in a scatter plot according to the first two principal components generated in the analysis by biometric indices for males (Figure 4) and females (Figure 5). Despite a different grouping pattern being visible across sexes, the consistent and interesting results obtained reinforce the suitability of the analysis and model generated.

Lastly, Table 7 exhibits the squared loadings of the different body measures in the analyses developed in each corporal region. The regions whose first two PCs yielded more than 70% of the cumulative explanatory variance were ‘Head’, in both sexes, and ‘Legs’, in females. Differences in the squared loadings of the biometric measures were obtained across sexes; however, most of the variables achieved their highest loadings in the first two PCs.

## 4. Discussion

The present work addresses, for the first time, the morphological diversity of the different turkey populations worldwide for which information was available. The lack of similar studies on this species, together with the small number of available papers and breeds described, highlights the secondary role this species plays in international research. Therefore, the results of the present work provide interesting conclusions, despite the limitations in the amount of available data and the lack of consistency in the parameter inclusion criteria between researchers. However, the high variance-explaining power of the first PCs, together with the consistent breed grouping in the spatial representation of the observations, demonstrates the suitability of this PCA approach. Therefore, this study proposes a new starting point in the study of the diversity of the domestic turkey.

The mean values for body weight and morphometric parameters suggest a large diversity between turkey breeds, especially for weight, wingspan, body length, and tarsus width variables. The present results also show a high sexual dimorphism in the species, which has been widely described in the bibliography [20,23,28,30,45]. However, a different effect of sexual dimorphism on biometric parameters across breeds could be suggested, as unselected genotypes show greater proportional differences between sexes. No similar reports have been made in turkeys since no studies comparing such a wide range of genotypes have been made before. However, this same conclusion was reached in a previous study analyzing carcass quality traits from different turkey breeds worldwide [26].

The correlation matrix shows different degrees of association between the biometric parameters, which seem to be stronger (more positive and negative values) in females. Skeletal and other size-related parameters were generally positively correlated among them, but were negatively correlated with beak width and snood length. Conversely, the lack of correlation between the weight of the bird and its breast circumference was surprising. These variables are generally described as having positive correlations in the literature [21,23,38,42,44,45,52]. However, these correlations are not always that strong in toms from indigenous breeds [38,40] and even negative correlations have been reported in local Nigerian hens [44]. This suggests that breast development varies greatly among breeds and that breast volume is not exclusively associated with body weight, at least in indigenous genotypes. For example, fast-growing strains are known to achieve great breast muscle yields; however, they have lost lung volume [53], which partially explains sudden deaths in handling and stressful stimuli. Conversely, local breeds maintain a proportionally greater respiratory capacity, possibly due to the demands of breeding in grazing systems [54,55]. In this context, poorer breast development and a greater leg mass in proportion to body size have been proposed as adaptations to pasturing systems in local turkey breeds [13]. Moreover, specific selection targets should also be considered. While the commercial industry demands broad-breasted turkeys for further processing, indigenous genotypes are commonly preferred for regional recipes, as are more harmoniously proportioned birds for oven cooking [56]. In addition to the genetic differences, the effects of the environment and management on the different genotypes should be considered. Some of the measured animals were reared in controlled environments with ad libitum feeding from hatching [13,30,42,43], while others were measured on their respective smallholder farms [21,38,40,41]. Therefore, the lack of correlation between body weight and breast circumference could be a consequence of including such a variety of genotypes and their different rearing systems.

In this line, there are striking breed differences in the STOCK index, which involves breast circumference and body length. This reinforces the idea that growth is not limited to the breast size for every genotype in the current analysis. However, despite a greater proportional breast development could be expected in the highly selected strains [57], the ‘Commercial’ group exhibited the lowest values for STOCK in the present study. However, the only two available reports of breast circumference in this genotype [18,42] did not specify the commercial strain, and a lighter line within the commercial hybrids could have been used, as the animals only weighed 8 kg at 6 months of age. Therefore, the results of the present index should be interpreted with caution.

Biometric indices generally showed negative correlations with size-related measures. This may be due to the fact that many of them act as divisors in some index estimation formulae. The strong correlation obtained between the MAS and BMI (0.963 in males and 0.877 in females) reflects that these two indices have almost the same coefficient, obtained from the same parameters. This indicates multicollinearity between these variables and the highly correlated body measures, suggesting that some of them should be excluded. In this line, the use of BMI over MAS in morphological studies could be suggested, since it has shown greater loadings in the analyses of both sexes.

A better PCA performance was obtained using only the biometric indices rather than all collected parameters. This is partly because the fewer variables included in the analysis, the simpler the model generated [58]. Biometric indices reflect the bird’s proportionality and harmony of the different body regions [21]. Therefore, analyzing breed variability using biometric indices may be particularly useful in breed characterization, as it combines information from different single body measures while reducing the model complexity through variable reduction. This good performance of the biometric indices in both breed diagnosis [13] and weight prediction [15] has been previously reported in turkeys and other poultry species [10,59].

Attending to the squared loadings of the biometric indices, LLEG was the first and third most explanatory variable in PC1 for males and females, respectively. Therefore, this index could be suggested as a highly breed-discriminating variable to be used in characterization studies. In this regard, the poultry meat industry has focused on breast yield due to Western consumers’ preference [60]. Therefore, the highly selected strains have undergone an exhaustive breast selection [57], losing, on the other hand, the proportional development of their legs [61]. By contrast, native and unselected genotypes might have maintained or even maximized their ancestral leg development. For example, there are commercial strains that have lost some of the primitive thigh bones [62]. Local breeds need sufficient leg development for locomotion in grazing and backyard systems [13], as well as for ancestral functions such as fighting and escaping from predators [55]. In addition, local consumer preferences may also have influenced the leg conformation of certain genotypes. This could be the case of the Eastern market, which prefers poultry meat from thighs and legs [60]. A similar effect could be expected for the local market, which demands the whole carcass and rejects broad-breasted turkeys [55,56].

TARS and SHAPE indices confirmed the hypothesis that the leg is an interesting area for breed traceability. Moderate (in males) to strong (in females) squared loadings were obtained for those indices in the second principal component (PC2). Since both indices are computed using tarsal measures (length, width, and depth), the shank is suggested as a highly recommended region for morphometric characterization studies. Although the shank has mainly been attached to sexual dimorphism [40,44], this corporal area is highly associated with other traits such as body length [38,40] and body weight [44,63]. In this way, the shank development served as a selection criterion for body mass gain during the 1980s [61]. On the other hand, the utility of the tarsus in breed diagnosis has been reported before [14,19]. The tarsus has also been considered the most important discriminator between two Italian breeds [13] and between Mexican subpopulations [23,52].

BMI and MAS were also among the most explanatory variables with high squared loadings in PC1. The fact that both indices are computed using live weight and body length suggests that these variables may play an important role in breed differentiation. These two were among the most variable parameters in Table 5 for both sexes, which, for body weight, has been widely reported in the literature [14,23,56]. On the other hand, there are contradictory reports on body length, which has been reported to have inconsistent explanatory potential in different turkey biometric studies [13,23]. In the present work, the differences between the Commercial and Zagorje genotypes are even three times higher, for both sexes. However, such differences could be due to the implementation of different anatomical references in the measurement, despite referencing the FAO’s guidelines in every case. On the other hand, the results may not be far from reality, as the Zagorje turkey could be one of the smallest turkey breeds in the world [35].

Interesting results can also be derived from the last PCA approach, which considered each corporal region separately. The head is the most informative region for males, as the PC1 yielded more than 90% of the explanatory variance, while none of the others achieved 70% in the first two PCs. In the case of females, the head region accounts for a significant explanatory potential as well, but is overtaken by the leg region. Therefore, the head is suggested as a region of great variability, despite the lack of results obtained using the skull index. In this regard, the direct association of the skull with vital functions such as predator detection or foraging may explain the wide range of shapes across populations [29]. Therefore, the lack of inclusion of head morphometric parameters in research studies is surprising. Only 12 articles involved in the present analysis included at least one of these variables. On the other hand, the importance of leg parameters in females reinforces the findings of the biometric analysis. This greater relative weight of female leg parameters in explaining the variance was also reported in the Nigerian turkey [44]. Therefore, the lack of results obtained in males could be due to the effect of sexual dimorphism. This is consistent with the studies included in the present analysis, which have reported significant differences between the sexes in leg morphometric parameters [13,22,40,42]. However, the presence of different levels of variability in body regions between sexes makes breeding management strategies difficult to implement. One suggestion would be to establish a breed pattern, with separate specifications for each sex. In this regard, different linear assessment criterion is usually followed in males and females in selection schemes [14]. However, this situation is more difficult to assess in local breeds, where there are generally limited resources or trained staff. Therefore, deeper studies should be conducted within each population to identify the characteristics that distinguish one breed from another.

Consistent breed grouping can be observed by attending to the spatial representation of observations, although different patterns were obtained between sexes. In the figure for males, a defined cluster involving the Bangladeshi, Ghanaian, and Tunisian turkeys is observed. Special mention must be made of the closeness between the Tunisian and Ghanaian turkeys, whose different feather varieties (white-black, red, and bronze) overlap, despite the geographical separation. The close similarity between northern and central Africa turkey landraces has been reported before in carcass and meat quality traits [26], which might suggest a common origin or expansion pathway in Africa. Females show another interesting cluster involving breeds from Croatia (Zagorje and Dalmatia), Italy (Nero d’Italia), and Tunisia. In this case, a possible ‘Mediterranean trunk’ could be suggested, as a consequence of the land and maritime connection from the first animals arrived in Spain from Mexico [64]. In this regard, molecular genetic studies have shown that some Mediterranean turkey populations (from Spain, Italy, and Egypt) are particularly genetically close compared to other turkey populations [65]. This supports the hypothesis of this Mediterranean trunk, possibly due to a historically intense maritime trade of animals. However, no molecular studies have included different turkey populations, and the suggested ‘African trunk’ is not supported by reports of molecular studies.

In both sexes, observations from the Norfolk Black and Mammoth Bronze breeds cluster close to the aforementioned native genotypes. Norfolk Black and Mammoth Bronze are unselected genotypes considered as ‘heritage’ landraces, ancestors of modern breeds [36,66]. Moreover, the Norfolk Black is also known as ‘Black Spanish’ [67], which would enhance the consistency of the Mediterranean breeds clustering. On the other hand, the present work involves observations of each genotype from two different research articles. In the case of females, observations were clustered together despite being obtained from different studies. In the case of males, observations were represented separately due to the effect of the PC2; however, observations were closely represented in the axis of the PC1. This reinforces the viability of the model since observations from the same breeds obtained from different studies were clustered together.

Finally, a wide variety of qualitative traits can be found in the studies involving the color and presence of cutaneous elements such as wattle, caruncles, or crown feathers. However, there are differences between the papers not only in the inclusion of these parameters but also in their nomenclature, which has led to their exclusion from the analysis. Among the main reasons for this could be the lack of specific guidelines for the morphometric characterization of turkeys, especially for qualitative traits. This is exacerbated by the general perception in the scientific community of a lack of phenotypic variability in the species compared to other older domesticated birds such as chickens [3]. Conversely, this theory is refuted when all of the above qualitative parameters are considered, together with new descriptions of special features in specific populations, such as the feather bun described in the ‘crested turkey’ [68]. Therefore, further studies are urgently needed on possible qualitative parameters that can be measured beyond feather color, which is only one trait encoded by a few genes [69]. This would lead to the development of common guidelines for the phenotypic characterization of turkey landraces worldwide.

## 5. Conclusions

The present study constitutes the first approach to the worldwide turkey morphological diversity. However, the recent nature of the scarce papers available included in the analysis highlights the increasing interest in the subject. Contrary to what has been suggested before, the present work proves the great morphological diversity in the domestic turkey. The biometric indices stand as an alternative to the individual measures in morphometric variability analyses, especially when the scarcity of data available limits discriminant analyses. BMI and LLEG in both sexes, and TARS and SHAPE in females, are highly suggested variables due to their explanatory potential in the PC1. On the other hand, MAS could be excluded from the analysis as this index showed redundancy and had a slightly lower explanatory ability. The importance of the leg in females was reinforced in the analysis made per body regions, while the head was the most explanatory region, particularly for males. The spatial representation of observations through the first two principal components suggested interesting groupings of breeds that could share common phylogenetic origin or migratory pathways. However, the limitations of the present study were still such a challenge, due to the heterogeneous inclusion criteria of variables and the scarce observations available. This situation was critical for the qualitative parameters, which were excluded from the analysis. Further studies with sufficient observations per breed and performing discriminant analyses are needed. These future studies could benefit from the information in the present work. Once the most variable body regions and the strong effect of sexual dimorphism have been highlighted, future researchers will be able to design and constitute a defined methodology and measurement protocol for homogeneous and comprehensive morphological studies in the species. This would lead to a deeper understanding of the association between variables and define those that are more important in breed diagnosis.

## Figures and Tables

**Figure 1 animals-15-02537-f001:**
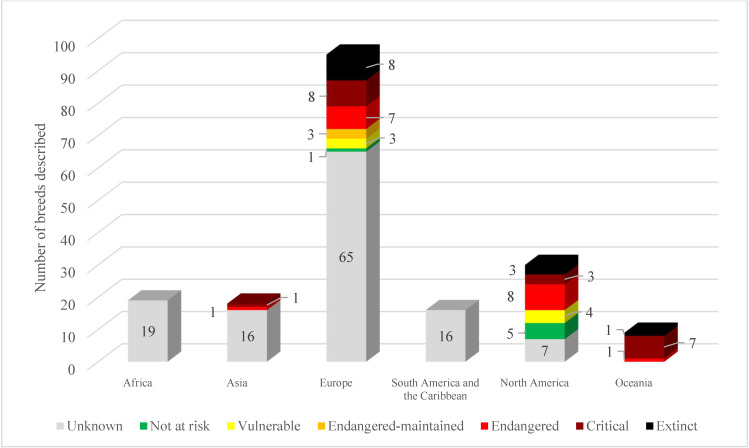
Detailed risk status of the world’s turkey breeds described in the DAD-IS database.

**Figure 2 animals-15-02537-f002:**
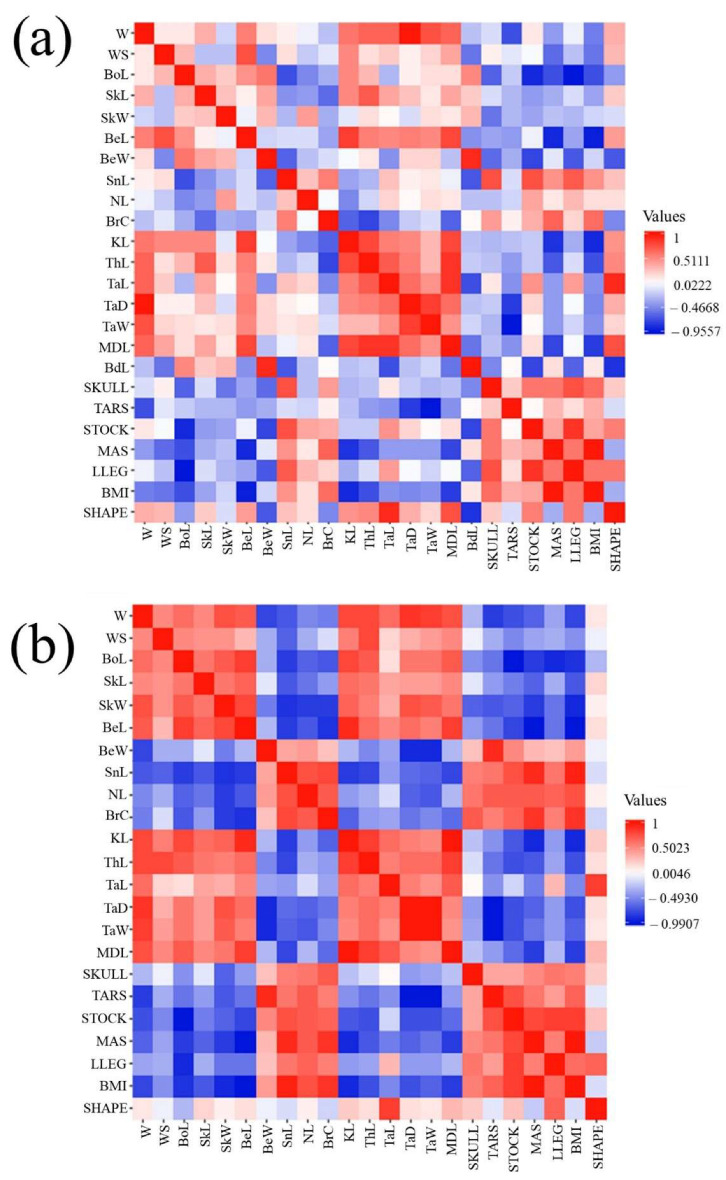
Territorial map depicting the correlation matrix between all parameters included in the analysis for males (**a**) and females (**b**). The abbreviations used were “W” for weight, “WS” wingspan, “BoL” body length, “SkL” skull length, “SkW” skull width, “BeL” beak length, “BeW” beak width, “SnL” snood length, “NL” neck length, “BrC” breast circumference, “KL” keel length, “ThL” thigh length, “TaL” tarsus length, “TaD” tarsus depth, “TaW” tarsus width, “MDL” middle toe length, “BdL” beard length, “SKULL” skull ratio, “TARS” tarsus ratio, “STOCK” stockiness, “MAS” massiveness, “LLEG” “long-leggedness”, “BMI” body mass index, and “SHAPE” shape index.

**Figure 3 animals-15-02537-f003:**
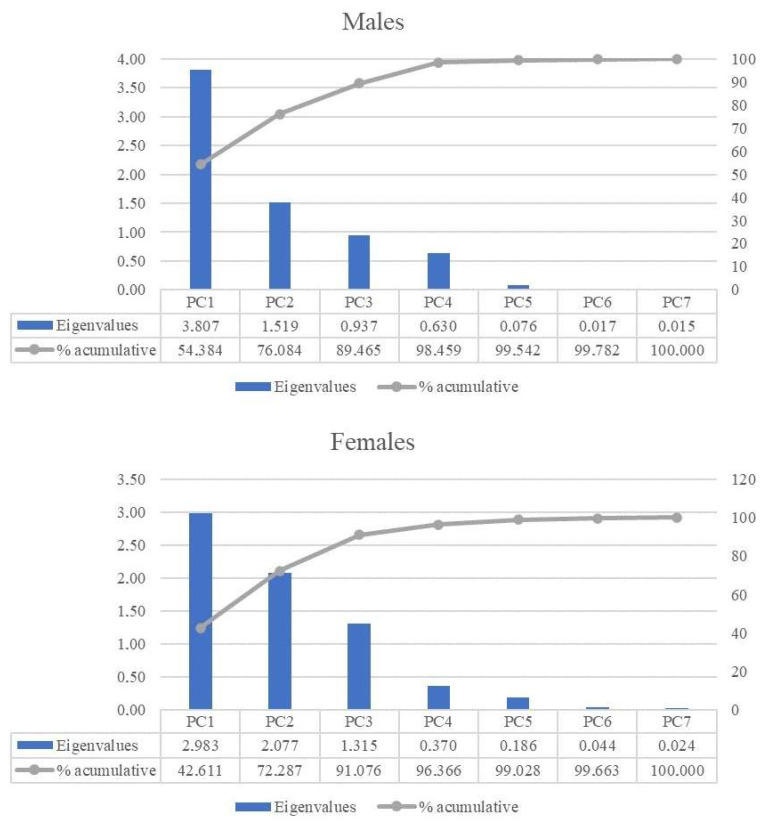
Eigenvalues and percentage of cumulative explanatory power (% accumulative) offered by the principal components (PCs) generated from the biometric indices analysis.

**Figure 4 animals-15-02537-f004:**
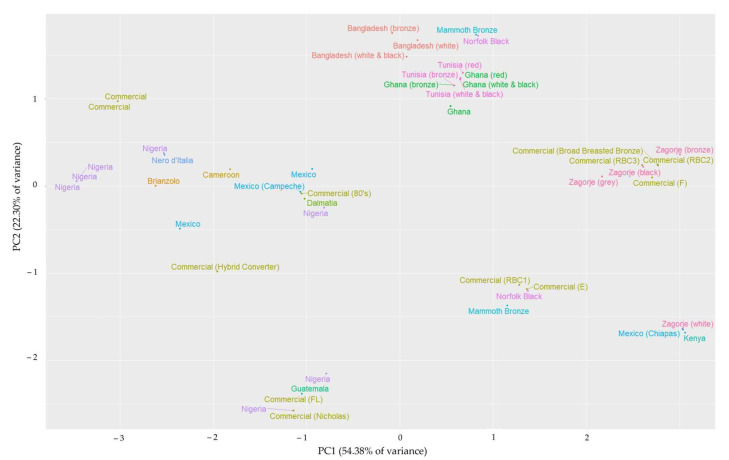
Spatial representation of male observations through the first two Principal Components generated in the biometric indices analysis (76.08% of accumulated variance). The tag color represents the 15 genotypes involved in the study. If available, the variety within the population is added by the bracket tags.

**Figure 5 animals-15-02537-f005:**
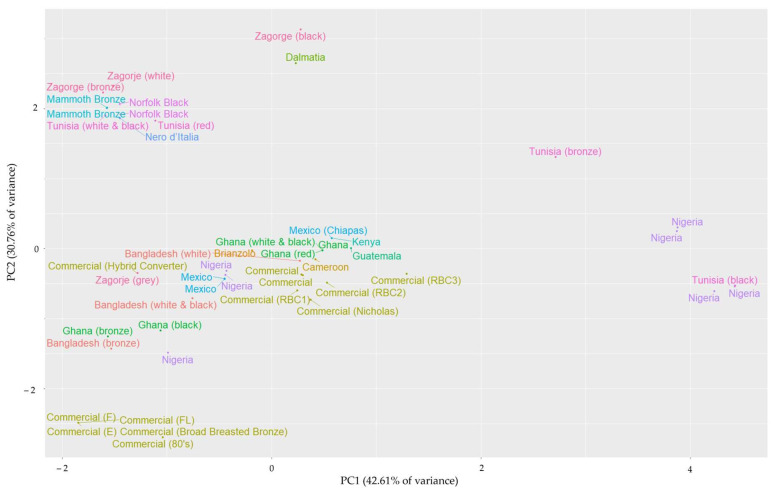
Spatial representation of female observations through the first two Principal Components generated in the biometric indices analysis (72.29% of accumulated variance). The tag color represents the 15 genotypes involved in the study. If available, the variety within the population is added by the bracket tags.

**Table 1 animals-15-02537-t001:** Name and area of distribution of the turkey genotypes included in the present analysis, and the reference from which the observations were collected.

Abbreviation	Genotype	Distribution	Measured Animals in Studies	References
COM	Commercial	Worldwide	1183 toms and 1149 hens	[14,20,29,30,31,32,33]
BRI	Brianzolo	Italy	17 toms and 29 hens	[13]
NDI	Nero d’Italia	Italy	17 toms and 29 hens	[13]
DAL	Dalmatian turkey	Croatia	10 toms and 20 hens	[34]
ZAG	Zagorje turkey	Croatia	20 toms and 80 hens	[35]
MAM	Mammoth Bronze	Worldwide	41 toms and 41 hens	[19,36]
NOR	Norfolk Black	Worldwide	41 toms and 41 hens	[19,36]
BAN	Local Bangladeshi	Bangladesh	25 toms and 25 hens	[37]
CAM	Local Cameroonian	Cameroon	95 toms and 141 hens	[38]
GHA	Local Ghanaian	Ghana	195 toms and 105 hens	[22]
GUA	Local Guatemalan	Guatemala	181 toms and 210 hens	[21]
KEN	Local Kenyan	Kenya	9 toms and 9 hens	[39]
MEX	Local Mexican	Mexico	526 toms and 332 hens	[16,23,40,41]
NIG	Local Nigerian	Nigerian	382 toms and 475 hens	[18,20,30,42,43,44,45]
TUN	Local Tunisian	Tunisia	93 toms and 108 hens	[46]

**Table 2 animals-15-02537-t002:** Abbreviations and formulas of estimation of each biometric index included in the present analysis, together with the reference from which the formula was obtained.

Abbreviation	Variable	Description
SkL	Skull length	Distance between the occiput to the tip of the beak
SkW	Skull width	Lateral distance between both zygomatic arches
BeL	Beak length	Distance from the base to the tip of the beak
BeW	Beak width	The maximum lateral distance, at the base of the beak
SnL	Snood length	Distance from the base to the tip of the snood
W	Weight	Bird’s live weight
WS	Wingspan	Distance between the tip of both extended wings
BoL	Body length	Distance between the beak’s tip and end of the non-feathered tail
NL	Neck length	Maximum distance between the occiput to the neck’s base
BrC	Breast circumference	Maximum thoracic perimeter through the keel and wings
KL	Keel length	Distance between both keel’s tips
ThL	Thigh length	Distance from the joint of the femur to the base of the shank
TaL	Tarsus length	Distance from the tibiotarsal joint to the beginning of the toe
TaD	Tarsus depth	Cranio-caudal distance at the middle of the metatarsal bone
TaW	Tarsus width	Lateral distance at the middle of the metatarsal bone
MDL	Middle toe length	Distance from the 3rd phalanx to the tip of the toe
SKULL	Skull ratio	Skull length/skull width
TARS	Tarsus ratio	Tarsus depth/tarsus width
STOCK	Stockiness	(Breast circumference/body length) × 100
MAS	Massiveness	(Weight/body length) × 100
LLEG	Long-leggedness	(Tarsus length/body length)100
BMI	Body mass index	Weight/(body length)^2^
SHAPE	Shape index	Tarsus length/^3^√(weight)

**Table 3 animals-15-02537-t003:** Mean values and standard deviation (if more than one report per genotype) of the morphometric parameters available for males.

	Commercial	Cameroonian	Kenyan	Dalmatian	Nero D’Italia	Zagorje	Tunisian	Ghanaian	Brianzolo	Bangladeshi	Norfolk Black	Mammoth Bronze	Mexican	Nigerian	Guatemalan
W (Kg)	10.96 ± 3.92	7.93 sr	7.20 sr	7.18 sr	6.76 sr	6.53 ± 0.51	6.25 ± 0.12	6.04 ± 0.47	5.47 sr	5.66 ± 0.15	5.62 ± 2.08	5.44 ± 2.46	5.36 ± 0.57	4.29 ± 1.70	3.69 sr
WS (cm)	80.27	-	-	-	60.38 sr	-	-	-	58.45 sr	-	-	-	69.80 sr	72.78 ± 0.15	-
BoL (cm)	65.48 ± 18.69	62.83 sr	-	-	63.31 sr	29.92 ± 0.91	34.58 ± 0.14	34.16 ± 0.92	62.91 sr	33.34 ± 1.16	33.85 ± 0.67	33.61 ± 1.48	49.74 ± 14.13	50.38 ± 13.90	55.01 sr
SkL (cm)	9.60 sr	-	-	11.35 sr	-	11.98 ± 0.35	-	7.20 sr	-	-	-	-	-	9.39 sr	11.25 sr
SkW (cm)	3.62 sr	-	-	4.35 sr	-	4.23 ± 0.17	-	-	-	-	-	-	-	-	4.26 sr
BeL (cm)	6.33 sr	-	-	-	-	3.64 ± 0.05	2.92 ± 0.01	-	-	5.48 ± 0.19	-	-	5.06 ± 0.15	5.19 ± 0.15	-
BeW (cm)	2.90 sr	-	-	-	-	-	-	2.86 ± 0.14	-	-	-	-	-	-	-
SnL (mm)	-	-	-	-	-	-	-	-	-	-	-	-	8.30 sr	-	7.38 sr
NL (cm)	25.00 sr	-	-	-	-	-	-	-	-	-	-	-	27.55 ± 1.73	23.72 ± 2.54	-
BrC (cm)	60.20 sr	62.02 sr	-	-	55.00 sr	-	-	-	53.62 sr	-	53.78 ± 8.79	52.14 ± 12.05	53.23 ± 7.93	47.35 ± 8.86	-
KL (cm)	19.70 ± 2.12	-	-	16.65 sr	17.66 sr	16.27 ± 0.75	14.63 ± 0.43	14.38	15.50 sr	-	-	-	16.29 sr	16.52 ± 2.76	-
ThL (cm)	23.47 ± 2.89	-	-	21.10 sr	-	22.16 ± 0.99	22.15 ± 0.42	21.80 ± 0.86	-	16.69 ± 0.56	20.31 sr	20.29 sr	15.07 sr	18.96 ± 4.50	-
TaL (cm)	17.74 ± 4.32	14.57 sr	16.9 sr4	15.77 sr	12.08 sr	15.70 ± 0.57	13.45 ± 0.03	13.40 ± 0.13	11.56 sr	11.06 ± 0.39	14.8 ± 2.54	14.53 ± 2.17	14.14 ± 0.84	11.34 ± 2.25	-
TaD (mm)	22.75 ± 3.17	-	-	-	-	-	-	-	-	-	-	-	-	-	-
TaW (cm)	15.97 ± 4.19	25.40 sr	-	-	10.32 sr	-	-	-	10.98 sr	-	-	-	15.35 ± 4.59	-	-
MDL (cm)	9.95 sr	-	-	-	-	-	-	-	-	-	-	-	-	8.02 sr	-
BdL (cm)	-	10.35 sr	-	-	-	-	-	-	-	-	-	-	6.83 sr	-	-

Where “W” represents weight, “WS” wingspan, “BoL” body length, “SkL” skull length, “SkW” skull width, “BeL” beak length, “BeW” beak width, “SnL” snood length, “NL” neck length, “BrC” breast circumference, “KL” keel length, “ThL” thigh length, “TaL” tarsus length, “TaD” tarsus depth, “TaW” tarsus width, “MDL” middle toe length, and “BdL” beard length. Cells with standard deviation represent averages estimated from more than one observation, while single reports have been identified with “sr”.

**Table 4 animals-15-02537-t004:** Mean values and standard deviation (if more than one report per genotype) of the morphometric parameters available for females.

	Commercial	Cameroonian	Kenyan	Dalmatian	Nero D’italia	Zagorje	Tunisian	Ghanaian	Brianzolo	Bangladeshi	Norfolk Black	Mamooth Bronze	Mexican	Nigerian	Guatemalan
W (Kg)	8.07 ± 2.75	4.89 sr	3.39 sr	4.26 sr	3.02 sr	3.85 ± 0.22	3.57 ± 0.11	3.69 ± 0.29	2.67 sr	4.01 ± 0.26	3.45 ± 0.92	3.36 ± 0.48	3.26 ± 0.05	3.346 ± 1.41	2.52 sr
WS (cm)	73.65 sr	-	-	-	49.71 sr	-	-	-	49.85 sr	-	-	-	58.40 sr	63.45 ± 0.92	-
BoL (cm)	59.15 ± 16.15	53.65 sr	-	-	53.59 sr	24.20 ± 0.78	28.66 ± 0.36	28.16 ± 0.74	51.40 sr	30.62 ± 0.69	29.84 ± 0.22	30.00 ± 0.28	45.51 ± 14.26	44.01 ± 12.47	46.73 sr
SkL (cm)	11.60 sr	-	-	9.62 sr	-	10.01 ± 0.19	-	4.76 sr	-	-	-	-	-	6.71 sr	9.99 sr
SkW (cm)	4.30 sr	-	-	3.68 sr	-	3.51 ± 0.04	-	-	-	-	-	-	-	-	3.74 sr
BeL (cm)	5.55 sr	-	-	-	-	3.30 ± 0.08	2.56 ± 0.11	-	-	4.36 ± 0.23	-	-	4.73 ± 0.34	4.79 ± 0.51	-
BeW (cm)	2.34 sr	-	-	-	-	-	-	2.50 ± 0.17	-	-	-	-	-	-	-
SnL (mm)	-	-	-	-	-	-	-	-	-	-	-	-	3.20 sr	-	1.95 sr
NL (cm)	22.92 sr	-	-	-	-	-	-	-	-	-	-	-	23.67 ± 1.47	18.93 ± 1.91	-
BrC (cm)	57.24 sr	48.70 sr	-	-	41.93 sr	-	-	-	41.02 sr	-	42.86 ± 1.61	42.24 ± 0.79	45.69 ± 11.56	41.51 ± 8.43	-
KL (cm)	17.89 ± 3.38	-	-	11.95 sr	12.50 sr	12.46 ± 0.27	11.49 ± 0.22	10.50 sr	12.00 sr	-	-	-	-	13.03 ± 1.66	-
ThL (cm)	21.07 ± 2.19	-	-	17.78 sr	-	18.27 ± 0.49	17.79 ± 0.54	17.87 ± 0.51	-	13.41 ± 0.36	16.96 sr	17.50 sr	13.01 sr	15.89 ± 3.66	-
TaL (cm)	14.54 ± 3.34	11.66 sr	13.50 sr	12.13 sr	9.63 sr	12.60 ± 0.35	10.64 ± 0.35	10.55 ± 0.37	9.59 sr	9.76 ± 0.22	12.16 ± 3.06	11.84 ± 2.61	11.71 ± 0.79	9.23 ± 1.99	-
TaD (mm)	20.25 ± 3.29	-	-	-	-	-	-	-	-	-	-	-	-	-	-
TaW (cm)	14.54 ± 4.09	25.20 sr	-	-	6.70 sr	-	-	-	7.58 sr	-	-	-	9.00 sr	-	-
MDL (cm)	8.17 sr	-	-	-	-	-	-	-	-	-	-	-	-	7.20 sr	-

Where “W” represents weight, “WS” wingspan, “BoL” body length, “SkL” skull length, “SkW” skull width, “BeL” beak length, “BeW” beak width, “SnL” snood length, “NL” neck length, “BrC” breast circumference, “KL” keel length, “ThL” thigh length, “TaL” tarsus length, “TaD” tarsus depth, “TaW” tarsus width, and “MDL” middle toe length. Cells with standard deviation represent averages estimated from more than one observation, while single reports have been identified with “s.r.”.

**Table 5 animals-15-02537-t005:** Mean values and standard deviation (if more than one report per genotype) of the biometric indices in both sexes.

		SKULL	TARS	STOCK	MASS	LLEG	BMI	SHAPE
COM	M	2.65 sr	1.43 ± 0.21	78.93 sr	11.59 sr	21.01 ± 13.76	0.101 ± 0.087	8.32 ± 1.95
	F	2.69 sr	1.38 ± 0.16	83.59 sr	11.91 sr	21.06 ± 15.46	0.116 ± 0.100	7.44 ± 1.84
CAM	M	-	-	98.71 sr	12.62 sr	23.19 sr	0.201 sr	7.31 sr
	F	-	-	90.77 sr	9.11 sr	21.73 sr	0.170 sr	6.87 sr
KEN	M	-	-	-	-	-	-	8.77 sr
	F	-	-	-	-	-	-	8.99 sr
DAL	M	2.61 sr	-	-	-	-	-	8.17 sr
	F	2.61 sr	-	-	-	-	-	7.48 sr
NDI	M	-	-	86.87 sr	10.68 sr	19.08 sr	0.168 sr	6.39 sr
	F	-	-	78.24 sr	5.64 sr	17.97 sr	0.110 sr	6.66 sr
ZAG	M	2.83 ± 0.03	-	-	21.84 ± 1.57	52.49 ± 2.38	0.730 ± 0.056	8.40 ± 0.27
	F	2.85 ± 0.05	-	-	15.90 ± 0.61	52.07 ± 0.66	0.657 ± 0.026	8.04 ± 0.12
TUN	M	-	-	-	18.09 ± 0.36	38.91 ± 0.11	0.523 ± 0.011	7.30 ± 0.06
	F	-	-	-	12.46 ± 0.54	37.15 ± 1.57	0.435 ± 0.024	6.96 ± 0.19
GHA	M	-	-	-	17.67 ± 0.98	39.23 ± 0.72	0.517 ± 0.017	7.36 ± 0.14
	F	-	-	-	13.14 ± 1.38	37.47 ± 1.02	0.468 ± 0.062	6.83 ± 0.34
BRI	M	-	-	85.23 sr	8.69 sr	18.38 sr	0.138 sr	6.56 sr
	F	-	-	79.81 sr	5.19 sr	18.66 sr	0.100 sr	6.91 sr
BAN	M	-	-	-	17.00 ± 0.33	33.19 ± 1.63	0.510 ± 0.026	6.20 ± 0.25
	F	-	-	-	13.09 ± 0.61	31.91 ± 1.41	0.427 ± 0.014	6.15 ± 0.27
NOR	M	-	-	158.64 ± 22.81	16.56 ± 5.83	43.79 ± 8.38	0.487 ± 0.162	8.54 ± 2.52
	F	-	-	143.61 ± 4.31	11.56 ± 3.01	40.80 ± 10.57	0.387 ± 0.098	8.20 ± 2.77
MAM	M	-	-	154.49 ± 29.02	16.03 ± 6.61	43.43 ± 8.38	0.473 ± 0.176	8.56 ± 2.58
	F	-	-	140.81 ± 3.97	11.20 ± 1.71	39.44 ± 8.32	0.374 ± 0.060	7.96 ± 2.12
MEX	M	-	-	103.25 ± 18.16	11.15 ± 2.46	28.89 ± 6.39	0.242 ± 0.098	8.08 ± 0.41
	F	-	-	89.49 ± 23.04	7.50 ± 2.47	25.93 ± 6.91	0.182 ± 0.111	7.89 ± 0.53
NIG	M	-	-	93.60 ± 40.93	7.18 ± 1.66	24.74 ± 11.51	0.128 ± 0.097	6.81 ± 1.41
	F	-	-	90.39 ± 35.64	6.25 ± 1.39	22.92 ± 11.02	0.125 ± 0.093	6.01 ± 1.31
GUA	M	2.64 sr	-	-	6.71 sr	-	0.122 sr	-
	F	2.67 sr	-	-	5.39 sr	-	0.12 sr	-

Cells with standard deviation represent averages estimated from more than one observation, while single reports have been identified with “s.r.”.

**Table 6 animals-15-02537-t006:** Squared loadings of each biometric index for the first five principal components.

		PC1	PC2	PC3	PC4	PC5
MALES	SKULL	**0.498**	0.168	0.250	0.058	0.026
	TARS	0.003	**0.489**	0.290	0.216	0.002
	STOCK	**0.471**	0.064	0.388	0.055	0.021
	MAS	**0.721**	0.211	0.001	0.048	0.013
	LLEG	**0.943**	0.032	0.000	0.014	0.000
	BMI	**0.825**	0.127	0.003	0.032	0.003
	SHAPE	0.345	**0.429**	0.005	0.207	0.012
		PC1	PC2	PC3	PC4	PC5
FEMALES	SKULL	0.458	0.002	**0.464**	0.003	0.067
	TARS	0.153	**0.588**	0.025	0.233	0.002
	STOCK	0.062	0.329	**0.548**	0.004	0.053
	MAS	**0.781**	0.027	0.127	0.036	0.012
	LLEG	**0.614**	0.347	0.004	0.010	0.006
	BMI	**0.892**	0.002	0.077	0.008	0.006
	SHAPE	0.023	**0.783**	0.071	0.077	0.040

The highest squared loadings for each variable are shown in bold. Cell color intensity increases with increasing loading values.

**Table 7 animals-15-02537-t007:** Squared loadings of each body measure for the first five principal components and their variance explanation accumulation.

		MALES					FEMALES				
		F1 (90.37%)	F2 (97.57%)	F3 (99.06%)	F4 (99.80%)	F5 (100%)	F1 (60.56%)	F2 (79.34%)	F3 (97.04%)	F4 (99.52%)	F3 (100%)
HEAD	SkL	**0.747**	0.245	0.006	0.002	0.000	**0.782**	0.052	0.109	0.054	0.003
	SkW	**0.931**	0.051	0.001	0.013	0.003	0.356	0.255	**0.385**	0.000	0.004
	BeL	**0.942**	0.001	0.057	0.000	0.000	**0.789**	0.141	0.005	0.063	0.002
	BeW	**0.923**	0.063	0.008	0.000	0.005	0.179	**0.487**	0.329	0.005	0.000
	SnL	**0.975**	0.000	0.002	0.021	0.002	**0.923**	0.004	0.057	0.001	0.015
		F1 (41.80%)	F2 (67.41%)	F3 (85.60%)	F4 (94.45%)	F5 (98.36%)	F1 (38.05%)	F2 (64.86%)	F3 (81.57%)	F4 (92.52%)	F5 (96.84%)
TORSO	W	0.109	0.075	**0.750**	0.053	0.002	**0.657**	0.079	0.083	0.079	0.056
	WS	**0.633**	0.081	0.002	0.252	0.019	**0.685**	0.132	0.079	0.000	0.009
	BoL	**0.633**	0.009	0.099	0.192	0.067	0.040	**0.400**	0.392	0.131	0.037
	NL	0.337	**0.520**	0.042	0.001	0.090	**0.554**	0.038	0.280	0.024	0.091
	BrC	**0.465**	0.327	0.143	0.022	0.000	0.088	**0.701**	0.090	0.028	0.068
	KL	0.332	**0.525**	0.055	0.010	0.057	0.260	0.259	0.078	**0.395**	0.000
		F1 (37.87%)	F2 (64.68%)	F3 (85.13%)	F4 (93.99%)	F5 (100%)	F1 (57.95%)	F2 (85.65%)	F3 (95.63%)	F4 (99.54%)	F5 (100%)
LEGS	ThL	0.172	**0.604**	0.038	0.184	0.002	**0.486**	0.170	0.340	0.004	0.000
	TaL	**0.734**	0.050	0.059	0.006	0.152	**0.710**	0.093	0.121	0.074	0.001
	TaD	**0.592**	0.179	0.073	0.035	0.121	**0.568**	0.396	0.000	0.027	0.009
	TaW	0.382	0.036	**0.435**	0.133	0.014	**0.554**	0.427	0.001	0.008	0.011
	MDL	0.013	**0.471**	0.418	0.086	0.012	**0.580**	0.299	0.037	0.082	0.002

The highest squared loadings for each variable are shown in bold. Cell color intensity increases with increasing loading values.

## Data Availability

The data used to support the findings of this study can be made available by the corresponding author upon request.

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
