# Peer review of "Explaining Global Turkey Biometric Diversity Through Principal Component Analysis"

_animals, 2025, doi:10.3390/ani15172537_

Round 1
Reviewer 1 Report
Comments and Suggestions for Authors
The present work addresses for the first time the morphological diversity of the different turkey populations worldwide. The Introduction chapter provides an overview of current knowledge on this research. However, please check whether the turkeys originated only from Meleagris gallopavo? Are the data from the DAD-IS database the latest? The Materials and Methods chapter lacks information on the calculation of Pearson correlations or a description of the scale for assessing the strength of correlations (whether the Guilford, Stanisz, or another scale). Is the measurement of tarsus length Or tibiotarsus length—birds have fused tarsal (heel) and metatarsal bones? This chapter also lacks information on how many birds the data came from. The Results chapter is well-written, but when describing correlations, the vocabulary for correlation strength should be used according to the scale used. In the discussion section, please indicate the use of body measurement in breeding and conservation flocks to maintain breed/strain-specific conformation traits and the relationship with muscle and carcass fatness when working on pedigree farms.
General comments
Please take the comments provided above into account when proofreading the article.
Please prepare the "references" section according to the Animals journal requirements (Abbreviated name journal name, long dash for page range).
Detailed comment
L33 thight length (see L153) Or leg length?
L49 and others square brackets [1] instead of regular brackets (1)
L93 Is reference [20] about chickens?
L108 and others [21,22], no spaces between references
Reviewer 2 Report
Comments and Suggestions for Authors
To Authors,
This research is useful and can be used as a good academic reference as the results reveal insights into explaining global turkey biometric diversity through principal 2 component analysis. However, there are some questions need to be completed.
- The analysis covers 15 breeds out of 187 described globally. How representative are these breeds for making conclusions about worldwide turkey diversity? Could regional underrepresentation or uneven sample distribution bias the PCA results?
- The study excludes qualitative variables (plumage, wattles, etc.) due to inconsistent reporting. However, these traits may hold important discriminatory power. Could the authors propose a framework for harmonizing qualitative descriptors so that they may be incorporated into future global analyses?
- The authors found negative associations between breast circumference and body weight, which contradict prior studies. Could these discrepancies be due to measurement inconsistencies, genotype-environment interactions, or the pooling of heterogeneous data sources? Further explanation is needed.
- The paper suggests “African” and “Mediterranean” trunks of turkey diversity. While intriguing, how do the authors reconcile these morphometric groupings with existing genomic evidence? Can morphometric PCA alone reliably support hypotheses about migration routes or phylogenetic origin?
- Since different traits are discriminant in males and females, how might this affect conservation strategies, breed characterization, and breeding programs where only one sex is typically measured or prioritized? This point deserves further elaboration.
- Observations included birds aged 4–11 months. Could this age spread introduce confounding effects on morphometric variability, particularly for breeds with slower growth?
- Some measurements were taken using different anatomical references. How did the authors ensure consistency beyond referencing FAO guidelines? Were sensitivity analyses performed to test robustness?
- The authors set a 70% variance-explained threshold. Could they justify this choice in the context of high-dimensional morphometric data, where lower thresholds are sometimes acceptable?
Reviewer 3 Report
Comments and Suggestions for Authors
The topic addressed in this paper is interesting and relevant for understanding the global biometric diversity of turkeys, however, the study requires several methodological and clarity improvements in the presentation of results before it can be considered for publication.
! More emphasis should be placed on explaining why this research is important. For example, highlighting its relevance in the context of turkey meat production, from the perspective of both consumers and producers, would strengthen the manuscript.
! It would be valuable to include a separate subsection focusing on the genetic diversity of turkeys, as well as on the phenotypic characteristics analyzed according to specific factors such as breed, sex, and the climate of the rearing area.
Introduction:
- Attention: Bibliographic references are not cited in the text in square brackets as required in the journal template...review the entire manuscript! g., [1] or [2,3], or [4–6] not (1) or (2,3), or (4–6) !
- Please note that when abbreviations are used, their full meaning should be provided at first mention in the text.. example in introduction-line 53,64 etc…. review the entire manuscript.
- I suggest that Figure 1 be moved from the Introduction section to a more appropriate part of the manuscript. Additionally, the values should be added to the chart to enhance clarity and improve visual interpretation
- Clearly emphasize at the end of the introduction what the purpose of this research is and the degree of novelty of the study!
Materials and Methods
- Line 109.111,161…cite the reference...do not write the full website..it is added at the end in the References section!
- For each analysis program/software used, please provide the name of the software, its version number, and the developer or provider. Just example: SPSS Statistics 25.0 (Armonk, NY, USA)
Results
- In Figure 2, explain the abbreviations used in the subtitle for better clarity and reader understanding…
- Also for figure 3, explain that means PC1, PC2 etc under the figure!
- In Figures 4 and 5, colors are used, please clarify what they represent. It might be helpful to add a legend to these figures for better understanding?
- In Table 4, different shades of green are used, clarify what they represent.
Conclusions
I recommend that the Conclusions section highlight the practical implications and potential applications of the findings presented in this study.
Reviewer 4 Report
Comments and Suggestions for Authors
Clarity & consistency
-
Specify exact search date range and filters (language, year).
-
Briefly explain how you reduced repository bias (since PubMed/Web of Science weren’t used).
-
Clarify age definition (how “adult weight” was determined).
-
Add examples of anatomical landmarks for body length, etc.
-
Flag MAS as optional since it overlaps with BMI.
-
For Commercial STOCK anomaly, mark results as “cautious interpretation.”
Presentation tweaks
-
Add an abbreviation table (W, WS, BoL, MFL, etc.).
-
Correct MFL wording → “middle toe length,” not “finger.”
-
Table captions (esp. Tables 3–5): explain whether values are pooled means, study means, or single-source.
-
Add “(single report)” note where no SDs are shown.
-
Figures 3–5: include % variance (PC1+PC2) in captions.
-
(Optional) Add ellipses/convex hulls in PCA plots for better group visualization.
-
Proofread for spacing/units consistency (mm, cm, ±, etc.).
Interpretation tightening
-
Acknowledge why breast–size correlations differ from earlier single-population studies (heterogeneity).
-
Add 1 line on why leg traits are stronger in females (dimorphism).
-
Suggest standardizing head landmarks since male PCA loads heavily on head measures.
Housekeeping
-
Replace placeholders: “Animals 2025, 15, x … doi: 10.3390/xxxxx,” received/revised dates, and Supplementary link.
-
Strengthen data availability (share harmonized dataset if possible).
-
Ethics: clarify “data from published studies, no new animal handling.”
Spot/line-level suggestions
-
Title: “Explaining global turkey biometric diversity with principal component analysis” (shorter).
-
Simple Summary: Trim 1–2 sentences for non-specialists.
-
Methods: List XLSTAT version and R/ggplot2 version used.
-
Tables 3–4: Mark single-value cells as “from single report.”
May be improved with the help of professionals.
Round 2
Reviewer 2 Report
Comments and Suggestions for Authors
no comments
Reviewer 3 Report
Comments and Suggestions for Authors
I agree with the modifications made by the authors in accordance with the suggestions for improving the manuscript. The revisions adequately address the previous comments, and the adjustments enhance the clarity, coherence, and scientific rigor of the work. Also, the correction of formatting issues have strengthened the academic value of the manuscript.